# Future changes in the trading of virtual water

Neal T. Graham [1,2,3✉], Mohamad I. Hejazi[1,3], Son H. Kim[1], Evan G. R. Davies[4], James A. Edmonds [1] & Fernando Miralles-Wilhelm[1,2,3,5]

Water stressed regions rely heavily on the import of water-intensive goods to offset insufficient food production driven by socioeconomic and environmental factors. The water embedded in these traded commodities, virtual water, has received increasing interest in the scientific community. However, comprehensive future projections of virtual water trading remain absent. Here we show, for the first time, changes over the 21st century in the amount of various water types required to meet international agricultural demands. Accounting for evolution in socioeconomic and climatic conditions, we estimate future interregional virtual water trading and find trading of renewable water sources may triple by 2100 while non-renewable groundwater trading may at least double. Basins in North America, and the La Plata and Nile Rivers are found to contribute extensively to virtual water exports, while much of Africa, India, and the Middle East relies heavily on virtual water imports by the end of the century.

[1] Joint Global Change Research Institute, Pacific Northwest National Laboratory, College Park, MD, USA. [2] Department of Atmospheric and Oceanic Sciences, University of Maryland, College Park, MD, USA. [3] Earth System Science Interdisciplinary Center, College Park, MD, USA. [4] Department of Civil and Environmental Engineering, University of Alberta, Edmonton, AB, Canada. [5] The Nature Conservancy, Arlington, VA, USA. ✉email: neal.graham@pnnl.gov

Virtual water trade (VWT) is the amount of water, either green (soil moisture) or blue (renewable and nonrenewable), that is consumed in the production of agricultural goods that are then traded in the international market[1]. This trading acts to alleviate stresses in several water stressed regions[2–4]. Potential water savings associated with, and analyses of past virtual water trading have received increased attention in the research community[5–13] as future water stresses may leave some regions unable to meet their agricultural demands through domestic production alone[5,8]. These stresses can be driven by future socioeconomic conditions which are expected to cause a relative increase in global VWT[14,15]. While much of the water traded globally is green water or renewable blue water, recent studies have found increasing extraction of nonrenewable groundwater from deep aquifers to grow crops[16–20] that are then traded internationally[21–24]. Continued depletion and over-exploitation of nonrenewable groundwater has significant negative effects both regionally and globally, including, but not limited to, land subsidence, eventual sea-level rise, and water quality degradation[17,25,26]. However, the spatial and temporal characteristics of future all sources of VWT are generally unknown. Further, although many analyses of VWT have attempted to reconstruct the historical evolution of the virtual water trade network[2,4,8,27] and have concluded that VWT doubled between 1986 and 2007[6,8], assessments of future virtual water trading remain absent from the current literature.

Here we have found and quantified increases in global VWT throughout the century, using the Global Change Analysis Model (GCAM), a market equilibrium model that links socioeconomics, climate, water, energy, and land systems (Methods), and a business-as-usual scenario combination of Shared Socioeconomic Pathway 2 (SSP2) and Representative Concentration Pathway 6.0 (RCP 6.0)[28]. Accounting for future human and climate influences, VWT is shown to increase throughout the century for all water types. Exports of virtual water are found to originate from select basins around the world while showing a dependence on socioeconomic changes throughout the century, particularly population dynamics in China.

## Results

### Future water exports are concentrated in particular regions.
In total, virtual green water exports and virtual blue water exports more than triple from 905 billion $m^3$ and 56 billion $m^3$ in 2010 to more than 3200 billion $m^3$ and 170 billion $m^3$, respectively, by the end of the century in response to increases in population and the resultant demand increases (Fig. 1). Uncertainties in projected values of virtual water exports increase because of differences in climate impacts, as simulated by general circulation models (GCM) for the RCP6.0 scenario (Methods). Virtual nonrenewable groundwater exports at least double by the end of the century. Comparisons to previous global estimates (Table 1) show differences in virtual water trading intensities as a result of different trade data products and resultant aggregation of crop products. This study includes direct agricultural products, prior to processing, whereas other studies may include aggregate crop textiles and by-products, thus increasing trading values.

A large proportion of the virtual green water trade in 2010 is associated with oil crops (e.g. soybeans), a result consistent with previous studies[29] (Supplementary Fig. 4). Increases in corn, wheat, and oil crops and lead to significant virtual green export increases by 2100 (Fig. 2a). These three crop commodities represent the largest proportion of current VWT and the highest green to blue water ratio required for production[30]. Africa, Europe, and India represent the largest importers of virtual green water.

Virtual blue water trading shows significant differences arising in China, Pakistan, India, and the Middle East as the availability of water for irrigation decreases and populations change throughout the century[31] (Fig. 2b; Supplementary Figs. 4–7). In 2100, globally, China represents a large source of virtual water exports due to the trading of wheat and rice products (Fig. 2b). Interestingly, China shifts from importer currently[6,8,11] to exporter in the future[12], because of a reduced growth rate after 2030 that ultimately causes population to decline[31]. Reduced domestic demands allow the use of all excess production to meet international agricultural demands. Regions in Africa experience nearly the opposite effect as population rapidly increases throughout the century, resulting in increasing demand that is unmet by domestic production (Supplementary Fig. 3). The United States represents another main source of future virtual blue exports through corn, fibers, and oil crops, with a corresponding import of only miscellaneous crops (MiscCrops, e.g. fruits, vegetables, nuts), as part of the southwestern United States shifts production away from MiscCrops toward the end of the century. Finally, we have found an intensification of VWT in the early part of the century as population growth continues and exports originate from water-intensive regions of the Middle East, Pakistan, and India, while toward the end of the century, exports come from water-rich areas that require smaller amounts of water to grow (Supplementary Fig. 2). This shifting of global food production accounts for demand changes, water scarcity changes, and groundwater depletion that together result in an inability to meet demands from domestic production alone, consistent with previous studies[16,32,33].

Our results show a fivefold increase in virtual nonrenewable groundwater trade by mid-century, with an end-of-century value doubling that of 2010. The export of nonrenewable groundwater to meet international demands is concentrated in several main regions: The United States, Mexico, western South America, and northern Africa. On a temporal scale, water scarce regions export nonrenewable groundwater early in the century but cease to do so after mid-century as demand changes and groundwater depletion worsens (Supplementary Fig. 2).

### Only a few basins address global agricultural deficits.
Analysis of GCAM results at the 235 water-basin scale (Methods) permits for the identification of specific locations where virtual water exports originate. Comparing the 2050–2100 values of virtual green and blue exports (Fig. 3a–d) reveals an intensification of exports in most Chinese river basins. Blue and green exports are also concentrated in the Missouri River basin, the La Plata basin in South America, and the Murray-Darling basin in Australia. Each of these basins contains significant agricultural production and their water supplies will be used heavily to meet future demands.

Tracing the virtual nonrenewable groundwater exports over time shows an important evolution as basins in Saudi Arabia and the Indus River basin export substantial volumes in 2050, but do not contribute to the global nonrenewable groundwater exports in 2100. This is because extraction in the first half of the century from the large underground aquifers in Saudi Arabia and India[34,35] causes additional pumping to become too expensive to sustain in these regions.

The High Plains, Central Valley, and Mississippi Embayment aquifers are the three most over-exploited aquifers in the United States currently, particularly for irrigated agriculture[20,36], and our results show this trend continues through 2100 in terms of virtual nonrenewable groundwater exports. Exports are not shown from the Missouri River basin, on top of the deepest portion of the High Plains aquifer since groundwater recharge is greater than

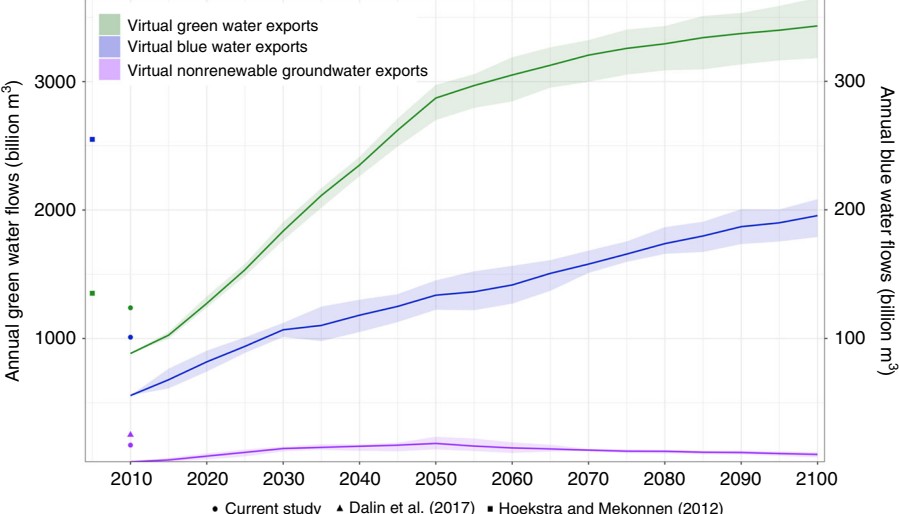

**Fig. 1 Annual water flows of green, blue, and groundwater embedded in agricultural trade for SSP2-RCP6.0.** Range of virtual green and blue water exports and the amount of nonrenewable groundwater depletion embedded in agricultural trade for all SSP2-RCP6.0 scenarios ($n = 6$ total scenarios), including effects of GCM uncertainty. Virtual green exports are shown on the primary y-axis (left) while virtual blue and nonrenewable groundwater exports are represented on the secondary y-axis (right). Solid lines represent the average for each water flow in SSP2-RCP6.0, while ribbons depict the full range of GCMs for the RCP6.0 scenario. Previous estimates of virtual water trade are shown as points and expanded upon in Table 1. Current study values are based on FAO country-level trade in 2010, all future estimates are between 32 GCAM regions.

**Table 1 Global physical water flows with comparisons to previous historical studies[7,21].**

| Water flows | Annual flows (billion m[c] per year) | | | | Source |
|---|---|---|---|---|---|
| | 1996–2005 | 2010 | 2050 | 2100 | |
| Virtual green exports | 1352 | | | | Hoekstra and Mekonnen[7] |
| | | 1239 | | | This study[a] |
| | | 905 | 2745–3040[b] | 3222–3708[b] | This study – SSP2-RCP6.0[c] |
| Virtual blue exports | 255 | | | | Hoekstra and Mekonnen[7] |
| | | 101 | | | This study[a] |
| | | 56 | 122–145[b] | 179–208[b] | This study – SSP2-RCP6.0[c] |
| Virtual nonrenewable groundwater exports | | 25 | | | Dalin et al.[21] [d] |
| | | 17 | | | This study[a] |
| | | 4 | 13.5–23.5[b] | 7.5–11.5[b] | This study – SSP2-RCP6.0[c] |

[a]Calculated using trade between each country from 2010 FAO country-level crop export data.
[b]Range across the five GCM suite of SSP2-RCP6.0 model runs.
[c]Calculated using trade between each of the 32 regions in GCAM. Does not include intraregional trade.
[d]Calculated using groundwater depletion rather than groundwater consumption.

nonrenewable extraction in this basin[37]; therefore, the groundwater withdrawn in the Missouri River basin is classified as renewable and is thus captured in the virtual blue exports. The Nile, La Plata, and Murray-Darling basins use nonrenewable groundwater to produce rice, fibers, and corn that is demanded outside of their regional boundaries (Supplementary Fig. 2C). These basins currently use significant amounts of groundwater for irrigated agriculture and have extensive, yet declining groundwater reserves[38–40].

## Discussion
Using GCAM to project the evolution of the global trade market, based upon changes in socioeconomic and climate conditions, provides a first assessment of changes in virtual water trade towards the end of the century, and addresses one of the major gaps identified in virtual water analyses[13]. In this analysis, we have built upon previous advances in the reconstruction of the historical global virtual water trade network[6,7,21,27] by linking potential future socioeconomic and climatic changes to

alterations in the production of agricultural goods, with the resulting price fluctuations in the global trade market causing a potential restructuring of global agricultural trading. Further, the work provides a first assessment of the quantities of nonrenewable groundwater extraction from aquifers around the world required to meet the international crop demand.

This study focused only on water used to grow agricultural crops, and while nearly 90% of blue water consumption is used for agricultural purposes[41], energy and industrial goods are also extensively traded in the global market; therefore, it is important to understand the international trade in these different sectors and how it may change into the future. Projecting such trade changes into the future is not trivial, but obtaining estimates based upon different climate and socioeconomic conditions can produce a wide range of potential trade development pathways. Further, although the main focus of this study has been on one socioeconomic scenario (SSP2) and one climate scenario (RCP6.0), it is important that future studies consider changes in the VWT network that result from different socioeconomic and climate conditions, and

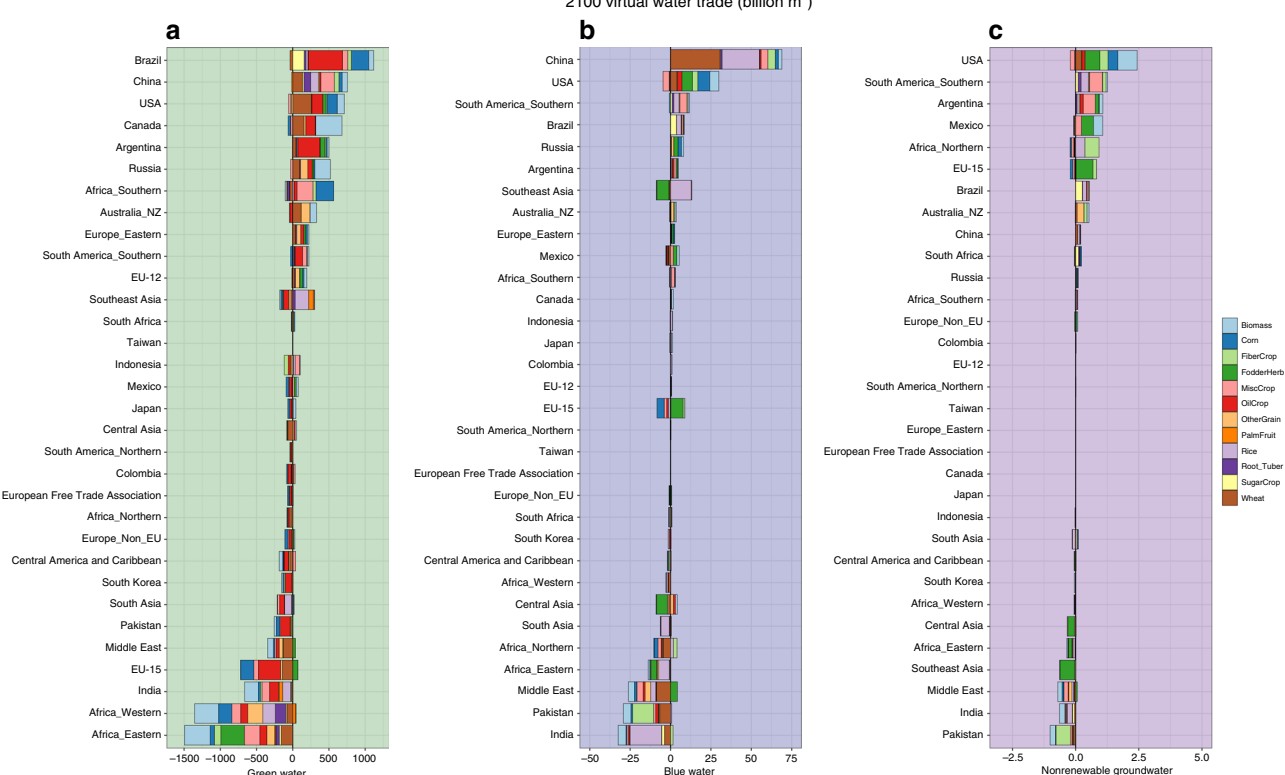

**Fig. 2 Virtual water trade fluxes by water type, region, and crop in 2100. a** Average global virtual green water trade (billion m$^3$), **b** virtual blue water trade (billion m$^3$), and **c** virtual nonrenewable groundwater trade (billion m$^3$) by crop and aggregate GCAM region in 2100 for all SSP2-RCP6.0 GCM scenarios ($n = 5$). Positive values represent exports while negative values imply virtual water imports.

that are likely to produce a larger range of potential outcomes. We provide an initial estimate of the uncertainty surrounding mid and end-of-century VWT values across a suite of SSP-RCP scenarios (Supplementary Figs. 8 and 9) and encourage further investigation to understand the drivers of change for various socioeconomic and climate mitigation scenarios.

## Methods

**Overall description.** This analysis uses GCAM to quantify the amount of water embedded in the global trading of agricultural goods. This water, called virtual water[1], is calculated based on how much water is consumed by the individual exported crop in the region where it was grown. In order to account for an evolving market and changing production conditions, we use a defined future socio-economic scenario, SSP2[28,42], matched with the RCP6.0 climate forcing scenario[43]. We introduce climate derived impacts from five general circulation models (GCM) to allow for changing water supply, crop yields, hydropower availability, and building energy demands (i.e., cooling and heating). We analyze the amount of green and blue water consumption that is embedded in global trade and differentiate between renewable surface and groundwater recharge, as well as nonrenewable groundwater to provide global estimates, regional contributions, and basin-level usage. Below we describe the GCAM model, scenario components, virtual water calculations in GCAM, and assumptions for the downscaling of exports and estimations of virtual water imports.

**GCAM.** GCAM is a market equilibrium model that links energy, water, land, economy, and climate systems[44–46]. GCAM adjusts prices of goods and services within each model time step to equilibrate the supply and demand of goods and services at each time step, and thus simultaneously clears markets across sectors. This study accounts for a limited supply of water by employing cost resource curves across all 235 basins that follow a logit formulation to determine the share of each water source (renewable, nonrenewable, and desalinated water) needed to meet the water demands within all basins[47,48]. As depletion of various water sources increases, the extraction price increases, which leads to compounding price increases in the goods and services that require higher-priced water sources.

Agricultural production in GCAM is computed endogenously by accounting for historical crop growth representations from MIRCA 2000 data in combination with yield estimates and a breakdown of irrigated and rainfed production. Water

consumption coefficients, both biophysical and blue water, are exogenous inputs in GCAM by country and crop type[7]. These are aggregated to the GCAM region scale for 12 crop types in GCAM, with two additional biomass crop-type water coefficients[49].

Agricultural trade within GCAM is modeled following a Heckscher–Ohlin method in which commodities are traded in a single global market where each region will see the same global price for that commodity. This allows each region to determine how much it will supply or demand of each commodity at that price. Using this method results in no preference for any region to demand certain commodities from another particular region.

**Key scenario components.** The SSP2 scenario, often referred to as a reference or business-as-usual scenario, represents a world with steady population growth through the middle of the century, at which time the global population begins to equilibrate toward a 2100 value of 9 billion people. Economic growth continues at present-day values, and thus fuel and energy preferences remain very similar to what they are today. For these reasons, this scenario represents one with medium challenges to both climate mitigation and adaptation[42]. Combining these socio-economic features with future climate changes, we implement a future RCP6.0 trajectory that results in end of century climatic forcing of 6.0 W/m$^2$. Quantitative assumptions for the SSP2 scenario are documented in separate studies[28,50].

**GCM derived climate impacts.** This study includes climatic impacts on water supply, agricultural productivity, hydropower availability, and building energy demands that are derived from five different general circulation models (GCM). We calculate each of these impacts by using downscaled and bias-corrected climate data from the Inter-Sectoral Impact Model Intercomparison Project (ISI-MIP)[51]. The global hydrologic model, Xanthos[52–54], calculates climate derived changes to renewable water supply at the GCAM 235-basin scale using necessary GCM outputs. Climate derived impacts to crop yield changes[55], hydropower availability[56], and building energy demands[57] are calculated from the same set of ISI-MIP models and the climate varying impacts are added to the SSP2 scenario. All datasets have been made publicly available for future use[58,59].

**Calculation of all virtual water components in GCAM.** Virtual water calculations in GCAM require several assumptions to account for its representation of trade as occurring across 32 regions, demands as regional, and production as basin level; further, the origins of imported goods are not traceable once exports are placed in the global market. In order to calculate the different components of virtual water

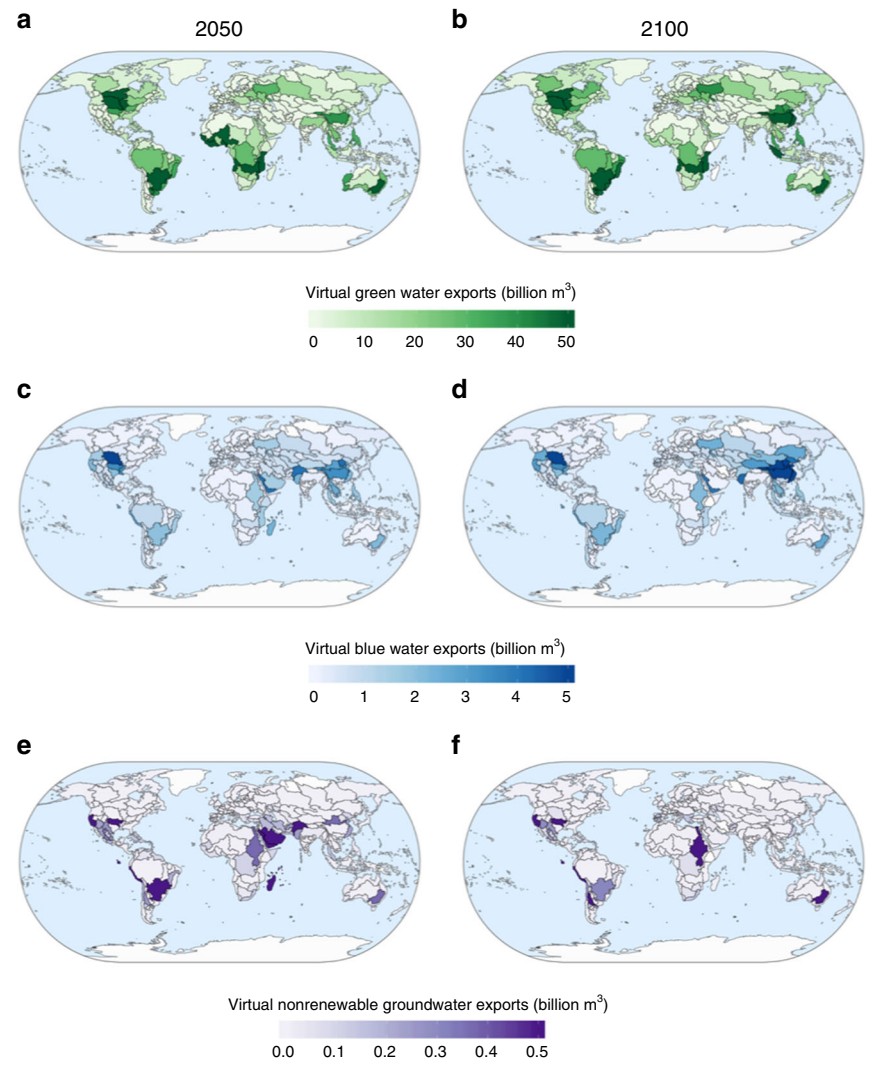

**Fig. 3 Basin-level virtual water exports in 2050 and 2100 for all sectors.** Virtual green water exports (billion m³) **a** and **b**, and blue water exports (billion m³), **c**, **d** in 2050 and 2100 respectively for the average of five GCM runs for SSP2-RCP6.0. Virtual nonrenewable groundwater exports (billion m³) in 2050 and 2100 for the same averaged GCM runs for SSP2-RCP.6.0 is shown in **e** and **f**. All values are considering the exports of agricultural crops only with additional, potentially necessary virtual water imports not considered.

trade, we must first calculate the regional and basin level trade. Although basin level imports are not calculated, all exports are trackable to the basin level, using the proportion of production as a proxy.

The second term in Eq. 1 takes the proportion of production, $P$, of any crop, $c$, and growth type, $g$, within a basin, $b$, to the total production of that crop in that region. This proportion is then multiplied by the regional demand, $D$, of that crop. This is due to GCAM modeling crop demands at the regional level. In order to scale this at the basin level we assume that the demand of a basin is proportional to the production from that basin. While demands are only modeled at the regional level, this is a good first-order approximation for estimating demands at a finer scale in GCAM. Growth types are classified as either rainfed, RFD, or irrigated, IRR and are determined endogenously within GCAM based upon the profitability of each crop type after the calibration period.

Once the proportion of regional demands is determined, it can then be subtracted from the basin level production, $P$, to determine the net surplus or deficit of a crop in a basin, $T$. Positive values of $T$, represent exports, $E$, whereas negative values represent the need for imports, $I$.

$$T_{b,c,g}(t) = P_{b,c,g} - \left[ D_C * \left( \frac{P_{b,c,g}}{\sum_{i=1}^{b} P_{b,c,g}} \right) \right] \quad (1)$$

Virtual green water exports, VGE, are calculated by considering the green water consumption, $GWC$, the basin level rainfed crop production, $P_{\text{RFD}}$, and the rainfed exports, $E_{\text{RFD}}$. Virtual green imports, VGI, must consider the amount of virtual green water that is in the global market summed over all regions, $r$, and basins, $b$, VGE, the ratio of imports, $I$, in a region, $r$, and total global imports of each crop. Finally, the total virtual green water trade (VGT) is calculated at the regional level

as the combination of the exports and imports of virtual green water.

$$\text{VGE}_{b,c}(t) = \frac{\text{GWC}_{b,c}}{\left( \frac{P_{b,c,\text{RFD}}}{\sum_{i=1}^{b} P_{b,c,\text{RFD}}} \right)} * E_{b,c,\text{RFD}} \quad (2A)$$

$$\text{VGI}_{r,c}(t) = \left( \sum_{i=1}^{b,r} \text{VGE}_{b,c} \right) * \frac{I_{r,c,\text{RFD}}}{\sum_{i=1}^{r} I_{r,c,\text{RFD}}} \quad (2B)$$

$$\text{VGT}_{r,c}(t) = \left( \sum_{i=1}^{b_r} \text{VGE}_{b,c} \right) + \text{VGI}_{r,c}(t) \quad (2C)$$

Virtual blue water analysis follows the same process as for green water, with the slight adjustment of accounting for irrigated production and trade, as well as the blue water consumption, BWC. Here virtual blue water exports (VBE), virtual blue water imports (VBI), and virtual blue water trade (VBT) require the production of irrigated agriculture.

$$\text{VBE}_{b,c}(t) = \frac{\text{BWC}_{b,c}}{\left( \frac{P_{b,c,\text{IRR}}}{\sum_{i=1}^{b} P_{b,c,\text{IRR}}} \right)} * E_{b,c,\text{IRR}} \quad (3A)$$

$$\text{VBI}_{r,c}(t) = \left( \sum_{i=1}^{b,r} \text{VBE}_{b,c} \right) * \frac{I_{r,c,\text{IRR}}}{\sum_{i=1}^{r} I_{r,c,\text{IRR}}} \quad (3B)$$

$$\text{VBT}_{r,c}(t) = \left(\sum_{i=1}^{b_r} \text{VBE}_{b,c}\right) + \text{VBI}_{r,c}(t) = 0 \qquad (3C)$$

Finally, the calculation of virtual groundwater exports (VGWE) considers the ratio of groundwater depletion in a basin, GWD, to the total blue water withdrawals in the basin, BWW. Multiplying this proportion by the virtual blue water exports yields the amount of the blue water exports that is from nonrenewable groundwater sources.

$$\text{VGWE}_{b,c}(t) = \text{VBE}_{b,c}(t) * \frac{\text{GWD}_b}{\text{BWW}_b} \qquad (4)$$

Total virtual groundwater trade (VGWT) and virtual groundwater imports (VGWI) are calculated in the same manner as Eq. 4, by considering the blue water imports and total trade as the first term on the right-hand side of the equation.

**Reporting summary**. Further information on research design is available in the Nature Research Reporting Summary linked to this article.

## Data availability
The data that support the findings of this study are available at https://doi.org/10.25584/1631593.

## Code availability
The processing code associated with this study is available at https://doi.org/10.5281/zenodo.3875735.

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

## Acknowledgements

This research was supported by the U.S. Department of Energy, Office of Science, as part of research in Multi Sector Dynamics, Earth and Environmental System Modeling Program. The Pacific Northwest National Laboratory is operated for DOE by Battelle Memorial Institute under contract DE-AC05-76RL01830. This research was also supported by the National Science Foundation Innovations at the Nexus of Food, Energy, and Water Systems under Grant EAR-1639327. Additionally, NOAA Award No. NA14NES4320003 for the "Cooperative Institute for Climate and Satellites".

## Author contributions

The project was designed by N.T.G., M.I.H., S.H.K., E.G.R.D., and F.M.W. Analytical work and data analysis was performed by N.T.G. The paper was written by N.T.G. with substantial revisions provided by M.I.H., S.H.K., E.G.R.D., J.A.E., and F.M.W.

## Competing interests

The authors declare no competing interests.
