## [Peer Review File · Nature Communications]

Reviewers' comments:

Reviewer #1 (Remarks to the Author):

TITLE: Future changes in the trading of virtual water

This study uses output from the GCAM model to make future projections of virtual water trade. GCAM is a market-equilibrium model that links energy, water, land, and climate. Output of the model allow the authors to determine how conditions of surplus and deficit will change by 2100 and predict the consequent reshaping of the global patterns of virtual water trade (VWT). Previous studies on this subject have either investigated the past spatiotemporal structure of VWT or developed empirical models of VWT that do not account for the effects of population growth, climate change and affluence. In this sense, this study is unique and I would be excited to see it published in Nat. Comm. There are, however, some major points that will need to be addressed. They will involve just some rewriting and reformatting of the figures (some of them are really hard to read).

MAJOR POINTS

- 1) Abstract: it is not clear that you will be looking at the effect of changes due to climate, population and economic development. Please, be clearer about it.
- 2) Throughout the manuscript you refer to GCAM as if it were something "universal" that everyone is expected to know about. Please define the acronym and explain in a sentence or two (already at the beginning of the paper) what this model is about, how it accounts for natural resources and how it is interfaced to climate change and socio-development scenarios.
- 3) Fig 1, Table 1: you refer to SSP and RCP scenarios without having introduced them, explained what they are and what they represent. Overall the manuscript has a lot of jargon. Why do you pick SSP2 and RCP6.0?
- 4) The way GCAM is used to calculate irrigation water consumption associated with crop production is a bit opaque. Please explain if you use a crop water model to determine the water footprint of crops in different regions of the world. How do you calculate the green and blue water footprints? Does the model account for plant physiological response to increased atmospheric CO₂ concentrations? Can you explain a bit better how you account for climate change scenarios in your calculations? You leave the reader with the impression that GCAM somehow does its magic.
- 5) How do you model trade? From each country/region to the rest of the world or do you model the actual flow of agricultural commodities and associated virtual water between country A and country B? How do you account for the dependence on population growth and dietary shifts?

MINOR POINTS

- 6) You seem to be concerned about groundwater depletion but with limited additional efforts you could also consider the impact on environmental flows (see Rosa et al., 2019);
- 7) Units and notation. "bcm" does not comply with the standard notation for the SI system. It could be used by some industries but it should not be used in a science journal. I suggest referring to Billion of m³ or $\times 10^9$ m³

8) Line 45: were do you look at virtual non-renewable groundwater exports? How?

9) Line 47: I would cite Mekonnen and Hoekstra 2012 in the figure caption and not here.

10) Figure 1: please add "green" and "blue" to the axes labels.

11) Line 103: I suggest citing here Konikow, GRL, 2011.

12) Figure 2: is difficult to read. There is no need for these figures to be on a ring: I suggest using a more standard linear structure and with a bigger bar chart for each of the three plots. This would allow you to have bigger labels. Also: some of the colors are way too similar. I suggest using also shadings with different patterns to make them more distinguishable.

13) Line 340: This is the first time you explicitly say that you account for climate change. Please, add a mention of climate change also in the abstract and introduction. Please, explain better if you account for the effect of climate change on evapotranspiration and irrigation water requirements and how.

14) Line 357: you refer to higher priced water sources. However, water is often subsidized? Do you account for the effect of subsidies on water consumption?

15) Please explain better eq 1

16) Lines 393-395: Did you account for the fact that there is green water consumption also in irrigated agriculture?

17) Eq 2: please explain what "br" stands for.

18) How do you determine the % of production that is rainfed?

Konikow, L. F. (2011). Contribution of global groundwater depletion since 1900 to sea-level rise. *Geophysical Research Letters*, 38, L17401.
<https://doi.org/10.1029/2011GL048604>

Rosa, et al.(2019). Global unsustainable virtual water flows in agricultural trade, *Environm. Res. Lett.*, 14, 114001.

Reviewer #2 (Remarks to the Author):

Major comments:

This manuscript claims that trading of renewable water sources may triple, and non-renewable groundwater trading may at least double by 2100. These findings have significant implications for future policy on global changes. There are several causes of uncertainty in the estimates. Is it possible to quantify the uncertainty? It would be useful to discuss how sensitive the findings are to small fluctuations in input parameters (socioeconomic factors, climatic factors, and agricultural crop growth factors). The authors are encouraged to address issues of sensitivity and uncertainty.

Minor comments:

Incorporate the contents of Table 1 into Figure 1 for more effective communication.

Shorten the captions of figures. If necessary, include key information in the text.

Reviewers' comments:

Reviewer #1 (Remarks to the Author):

TITLE: Future changes in the trading of virtual water

This study uses output from the GCAM model to make future projections of virtual water trade. GCAM is a market-equilibrium model that links energy, water, land, and climate. Output of the model allow the authors to determine how conditions of surplus and deficit will change by 2100 and predict the consequent reshaping of the global patterns of virtual water trade (VWT). Previous studies on this subject have either investigated the past spatiotemporal structure of VWT or developed empirical models of VWT that do not account for the effects of population growth, climate change and affluence. In this sense, this study is unique and I would be excited to see it published in Nat. Comm. There are, however, some major points that will need to be addressed. They will involve just some rewriting and reformatting of the figures (some of them are really hard to read).

MAJOR POINTS

1) Abstract: it is not clear that you will be looking at the effect of changes due to climate, population and economic development. Please, be clearer about it.

Thank you for this comment. We have updated the abstract to reflect the evolving socioeconomic and climate conditions within the scenarios analyzed.

“Here we show, for the first time, changes over the 21st century in the amount of various water types required to meet international agricultural demands. Accounting for evolution in socioeconomic and climatic conditions, we estimate future interregional virtual water trading across 235 global water basins and find trading of renewable water sources may triple by 2100 while nonrenewable groundwater trading may at least double.” Lines 18-23

2) Throughout the manuscript you refer to GCAM as if it were something “universal” that everyone is expected to know about. Please define the acronym and explain in a sentence or two (already at the beginning of the paper) what this model is about, how it accounts for natural resources and how it is interfaced to climate change and socio-development scenarios.

Thank you for this comment. We have adjusted the main text to provide the full name of GCAM as well as a brief explanation of the main capabilities of the model. Because of word limits, we have placed much of the model description in the Methods section after the main text and have expanded upon this further in the Supplementary Materials.

“Here we have found and quantified increases in global VWT throughout the century (Fig. 1 and Table 1, see Methods for scenario description), using the Global Change Analysis Model (GCAM) and a business-as-usual scenario combination of Shared Socioeconomic Pathway 2 (SSP2) and Representative Concentration Pathway 6.0 (RCP 6.0). GCAM is a market equilibrium model that links socioeconomics, climate, water, energy, and land systems (Methods)” Lines 40-45

3) Fig 1, Table 1: you refer to SSP and RCP scenarios without having introduced them, explained what they are and what they represent. Overall the manuscript has a lot of jargon. Why do you pick SSP2 and RCP6.0?

We have now introduced and defined both SSPs and RCPs in the GCAM introductory sentence while also providing a short justification for its inclusion. We have chosen an SSP2-RCP6.0 scenario as this is most reflective of a business-as-usual future. In order to avoid introducing significant driving forces that may have produced results that were skewed because of significant climate mitigation or socioeconomic development, we decided to choose the middle of the road development scenario. However, we have included a discussion and additional supplementary figures to illustrate the impacts of various socioeconomic development and climate future combinations. It can be seen that significant changes result from various combinations of socioeconomic and climate futures that are driven by slowing of population growth (SSP1), consistent population growth through the end of the century (SSP3), and significant investments in climate mitigation (RCP2.6). As such, the business-as-usual scenario was chosen as the main scenario to avoid these additional forcings on VWT.

“...a business-as-usual scenario combination of Shared Socioeconomic Pathway 2 (SSP2) and Representative Concentration Pathway 6.0 (RCP 6.0)” Lines 40-41

4) The way GCAM is used to calculate irrigation water consumption associated with crop production is a bit opaque. Please explain if you use a crop water model to determine the water footprint of crops in different regions of the world. How do you calculate the green and blue water footprints? Does the model account for plant physiological response to increased atmospheric CO₂ concentrations? Can you explain a bit better how you account for climate change scenarios in your calculations? You leave the reader with the impression that GCAM somehow does its magic.

Water footprints (blue and green) in GCAM are exogenous inputs from Mekonnen and Hoekstra (2010). The values are aggregated to the GCAM crop and region level to calculate water withdrawal coefficients. Using the same values from that study, we compare the regional water consumption of blue and green water in Supplementary Figures 11 and 12 which shows aggregation of their values across crops and countries has not significantly impacted the values.

The response to increased atmospheric CO₂ is captured in agricultural yield changes dependent on RCP scenario (Rosenzweig et al., 2014). These changes have been calculated exogenously, as changes in current harvested area weights, and are inputs to GCAM within the appropriate SSP-RCP scenario.

“Agricultural production in GCAM is computed endogenously by accounting for historical crop growth representations from MIRCA 2000 data in combination with yield estimates and a breakdown of irrigated and rainfed production. Water consumption coefficients, both biophysical and blue water, are exogenous inputs in GCAM by country and crop type⁷. These are aggregated to the GCAM region scale for twelve crop types in GCAM, with two additional biomass crop-type water coefficients.” Lines 368-373

5) How do you model trade? From each country/region to the rest of the world or do you model the actual flow of agricultural commodities and associated virtual water between country A and country B? How do you account for the dependence on population growth and dietary shifts?

GCAM models trade following the Heckscher-Ohlin method in which commodities such as agricultural crops are traded in a single global market where each region will see the same global price for that commodity. This allows each region to determine how much it will supply or demand of each commodity at that price. Using this method results in no preference for any region to demand certain commodities from another particular region. For these reasons we have been able to assess the exported virtual water as we know regions in which there is surplus production; however, the virtual water imports have been approximated by assuming that imports come proportionately from all exporters based on each individual region's volume of export by commodity.

Population growth is exogenous to GCAM, as are dietary patterns. This results in food demand within GCAM responding to increases in population or shifts in meat or vegetable/fruit consumption. The food demands in GCAM are calibrated to FAO statistics for the base year of 2010. They are then changed into the future based on the SSP storyline (explained further in Calvin et al., 2017). Agricultural demand is made up of three separate components: food demand, feed demand, and non-food demand. As population changes, these demands will change and thus influence the demands for agricultural goods within a region which then in turn alters the supply-demand balance of a region.

“Agricultural trade within GCAM is modeled following a Heckscher-Ohlin method in which commodities are traded in a single global market where each region will see the same global price for that commodity. This allows each region to determine how much it will supply

or demand of each commodity at that price. Using this method results in no preference for any region to demand certain commodities from another particular region.” Lines 374-378

MINOR POINTS

6) You seem to be concerned about groundwater depletion but with limited additional efforts you could also consider the impact on environmental flows (see Rosa et al., 2019);

Thank you for this comment. Environmental flows and environmental flow requirements are accounted for prior to water availability being added to GCAM. In the hydrologic model Xanthos, a certain percentage of water (in the case of GCAM, 25%) is considered unusable for environmental flow requirements in dry conditions (Turner et al., 2019). In addition, after the amount of “accessible water” is determined from the process previously noted, an additional reduction in the available amount of water is applied if groundwater depletion occurs in the basin where the new accessible water in the basin is determined to be demand-depletion/runoff.

Thank you for bringing this study to our attention, we have added this to the literature review for the manuscript.

7) Units and notation. “bcm” does not comply with the standard notation for the SI system. It could be used by some industries, but it should not be used in a science journal. I suggest referring to Billion of m^3 or $\times 10^9 m^3$

This has been changed throughout the text to billion m^3 .

8) Line 45: were do you look at virtual non-renewable groundwater exports? How?

In GCAM blue water resources are split between two different available sources, 1) renewable surface runoff and groundwater recharge and 2) non-renewable groundwater. Within each basin a certain percentage of water withdrawals comes from each of these sources which allows for the key assumption that ***each sector that extracts water from a particular basin does so in proportion to the renewable to non-renewable withdrawal ratio (right term in Equation 4)***. Therefore, the non-renewable groundwater exports represent a percentage of the virtual blue water exports of each basin, assuming that non-renewable groundwater depletion exists. Virtual non-renewable groundwater exports are explored throughout the manuscript, and additional clarifications regarding both calculations and assumptions have been added to the main body and methods sections.

9) Line 47: I would cite Mekonnen and Hoekstra 2012 in the figure caption and not here.

This has been changed.

10) Figure 1: please add “green” and “blue” to the axes labels.

This is been fixed for clarification.

11) Line 103: I suggest citing here Konikow, GRL, 2011.

This citation has been added.

12) Figure 2: is difficult to read. There is no need for these figures to be on a ring: I suggest using a more standard linear structure and with a bigger bar chart for each of the three plots. This would allow you to have bigger labels. Also: some of the colors are way too similar. I suggest using also shadings with different patterns to make them more distinguishable.

We have taken the reviewer's advice and have made the graphs vertical by traded water type. The color bar has also been changed to help differentiate between crop types.

13) Line 340: This is the first time you explicitly say that you account for climate change. Please, add a mention of climate change also in the abstract and introduction. Please, explain better if you account for the effect of climate change on evapotranspiration and irrigation water requirements and how.

Building on the response to Major Point #4, the effects of climate change on evapotranspiration and irrigated water requirements are reflected only in yield changes, as documented in Rosenzweig et al. (2014). GCAM does not have an embedded or coupled crop water model that allows for more comprehensive crop responses to climate changes. This point has been clarified in the text.

14) Line 357: you refer to higher priced water sources. However, water is often subsidized. Do you account for the effect of subsidies on water consumption?

In GCAM there is no cost for water consumption, rather the price is placed on water withdrawals. In line with previous estimations (Saglam, 2013), GCAM implements a price subsidy for irrigated water withdrawals which lowers the price by a factor of one hundred (Kim et al., 2016). This means that farmers will pay 1% of the price for water as other sectors in any basin. As water is continuously depleted, the price to use this water increases across all sectors to reflect additional measures needed to obtain this water (i.e. infrastructure changes to allow for increased groundwater pumping). While the price increases, the agricultural subsidy remains.

15) Please explain better eq 1

Equation 1 has some assumptions built in that, we agree, deserve further explanation. We can split equation 1 into two terms, $P_{b,c,g}$, and $\left[D_C * \left(\frac{P_{b,c,g}}{\sum_{l=1}^b P_{b,c,g}} \right) \right]$. The latter takes the proportion of production of any crop within a basin to the total production of that crop in that region. This

proportion is then multiplied by the regional demand of that crop. This is done because GCAM does not model basin level crop demands, rather regional crop demands. In order to scale this at the basin level we assume that the demand of a basin is proportional to the production from that basin. This has the potential to underestimate demands where there exists no production of certain crops while overestimating elsewhere due to disproportionate growing of crops across a region. While demands are only modeled at the regional level, this is a good first-order approximation of estimating demands at a finer scale in GCAM.

Once the portion of regional demands is determined, this value can then be subtracted from the basin level production to determine the net surplus or deficit of a crop in a basin.

This explanation has been updated in the manuscript.

“The second term in Equation 1 takes the proportion of production, P , of any crop, c , and growth type, g , within a basin, b , to the total production of that crop in that region. This proportion is then multiplied by the regional demand, D , of that crop. This is due to GCAM modeling crop demands at the regional level. In order to scale this at the basin level we assume that the demand of a basin is proportional to the production from that basin. While demands are only modeled at the regional level, this is a good first-order approximation of estimating demands at a finer scale in GCAM. Growth types are classified as either rainfed, RFD, or irrigated, IRR and are determined endogenously within GCAM based upon the profitability of each crop type after the calibration period.

Once the portion of regional demands is determined, this value can then be subtracted from the basin level production, P , to determine the net surplus or deficit of a crop in a basin, T . Positive values of T , represent exports, E , whereas negative values represent the need for imports, I .” Lines 540-552

16) Lines 393-395: Did you account for the fact that there is green water consumption also in irrigated agriculture?

Yes, this is accounted for in GCAM. In GCAM, it is assumed that irrigated agriculture partially meets water requirements from soil moisture (green water) and then all unmet water demands are met by blue water. Rainfed agriculture on the other hand consumes only green water. (See Chaturvedi et al. (2015) for future discussion on irrigated agriculture in GCAM). We have added this note to the main text of the manuscript to clarify to future readers. We have also updated the equations for virtual green water to better represent this fact.

However, the values of virtual green water trade do not change due to this equation change as the proportion of exports to production remains the same for irrigated and rainfed for this study and is expanded upon in the proof below for reference.

17) Eq 2: please explain what “br” stands for.

For all equations b_r represents all basins, b , in region, r . Therefore, when calculating virtual green water imports, we consider all virtual water exports of a particular crop from all basins in a region. This has been updated in the manuscript for clarified communication.

“Virtual green imports, VGI, must consider the amount of virtual green water that is in the global market summed over all regions, r , and basins, b , VGE, the ratio of imports, I , in a region, r , and total global imports of each crop.” Lines 423-425

18) How do you determine the % of production that is rainfed?

Crop production in GCAM is split between irrigated and rainfed and therefore the percent of rainfed in any basin is the proportion of rainfed to total production. Please see the figure below for a more detailed nested representation of crop production in GCAM. Note: Crops are separated between irrigated (IRR) and rainfed (RFD) as well as high (hi) and low (lo) fertilizer application rates.

Reviewer #2 (Remarks to the Author):

Major comments:

This manuscript claims that trading of renewable water sources may triple, and non-renewable groundwater trading may at least double by 2100. These findings have significant implications for future policy on global changes. There are several causes of uncertainty in the estimates. Is it possible to quantify the uncertainty? It would be useful to discuss how sensitive the findings are to small fluctuations in input parameters (socioeconomic factors, climatic factors, and agricultural crop growth factors). The authors are encouraged to address issues of sensitivity and uncertainty.

We thank the reviewer for taking the time to provide valuable comments on our manuscript.

We provide a point-by-point response to each of the major and minor points listed below.

Sincerely,

Neal Graham, on behalf of all authors

We have taken the suggestion of sensitivity and uncertainty seriously and have expanded the analysis across the entire SSP-RCP scenario suite in the supplementary section. In Supplementary Figures 9 and 10 we provide the range of virtual water trade across all SSP-RCP combinations which allows for the comparisons across socioeconomic factors (different SSPs for similar RCPs) and climatic factors (different RCPs for similar SSPs), as suggested. Within each of the climate scenarios we have included 4 different drivers (described in Graham et al., 2020; and copied below), one of which is agricultural productivity changes due to differing RCPs. We have thus included this concurrently with climatic factors, instead of isolating agricultural crop growth factors individually.

“Socioeconomic and Climate Scenarios

This study employs the use of temporally varying socioeconomics and climate systems. These systems are represented by the Shared Socioeconomic Pathways (SSP), for socioeconomic change, in combination with the Representative Concentration Pathways (RCPs), for climatic change. The SSPs are a set of five future scenarios with varying changes to global population, the economy (Riahi et al., 2017), and land use (Popp et al., 2017), which were designed to explore varying degrees of challenges to climate change adaptation and mitigation (O’Neill et al. 2017). These scenarios have been defined, both qualitatively and quantitatively, using a set of global integrated human-Earth systems models, in which individual models have been used to produce a marker scenario (Riahi et al., 2017) for singular SSPs (Calvin et al., 2017; Fricko et al., 2017; Fujimori et al., 2017; Kriegler et al., 2017; van Vuuren et al., 2017) and provide uncertainty ranges across all other SSPs. These results provide alternative projections of the evolution of the economy, energy systems, land systems, emissions, and climate. Within GCAM, quantitative assumptions of the SSPs have been made for the economy, energy sector,

land use, agricultural sector (Calvin et al., 2017), and for the water sector (Graham et al., 2018). The RCPs are a set of four future greenhouse gas (GHG) emission pathways in which end-of-century radiative forcing approaches four-levels by altering future greenhouse gas emissions and by changing underlying socioeconomic projections (van Vuuren et al., 2011). The SSPs and RCPs are combined to form a set of future global change scenarios which allows comprehensive socioeconomic assumptions to be matched with future radiative forcing pathways to achieve future global warming targets; creating the next set of scenarios that will provide the basis for future Intergovernmental Panel on Climate Change (IPCC) assessments (O'Neill et al., 2016; Eyring et al., 2016). Each is then matched with their individual Shared Policy Assumptions (SPAs) in order to account for differences in adaptation and mitigation strategies across the SSPs (Kriegler et al., 2014; Calvin et al., 2017). These combinations create a set of 15 potential global futures which are varied six times, for five GCMs and one set in which climate impacts across all sectors are neglected to produce 90 total scenarios.

Derived Climate Impacts

This study accounts for four different climatic impacts: water supply, agricultural productivity, hydropower capacity changes, and building energy demands derived from five general circulation models (GCMs). We calculate the impact on each aspect at four different radiative forcing levels and apply these to the appropriate scenarios within the SSP-RCP scenario matrix. Future water supply is calculated by using bias-corrected precipitation and temperature data derived for four RCPs from five CMIP5 generation GCMs as part of the Inter-Sectoral Impact Model Intercomparison Project [ISI-MIP; Warszawski et al. (2014)]. These values are entered into the global hydrologic model Xanthos (Li et al. 2017, Liu et al., 2018; Vernon et al. 2019, Supplementary Material), which calculates accessible water, i.e. available surface runoff and groundwater recharge, at the GCAM 235-basin scale at five-year time steps. Climate derived impacts to crop yield changes (Rosenzweig et al. 2014), hydropower capacity changes (Turner et al., 2017), and building energy demands (Clarke et al., 2018) are calculated from the same set of ISI-MIP models and the climate varying impacts are added to their respective RCP scenarios. By including scenarios that both include and exclude climate impacts we can account for the compounding effects of changing hydrologic conditions, hydropower availability, crop yields, and energy demands on water scarcity, while also separating the impact of human activities from that of the climate system, which include changing water demands for agriculture, power generation, industry, and public supply.”

Minor comments:

Incorporate the contents of Table 1 into Figure 1 for more effective communication.

Thank you for this suggestion. This has been added for clarity.

Shorten the captions of figures. If necessary, include key information in the text.

Thank you for this comment. We have consolidated the figure captions as suggested.

References

Mekonnen, M. M., and Arjen Ysbert Hoekstra. "A global and high-resolution assessment of the green, blue and grey water footprint of wheat." *Hydrology and earth system sciences* 14, no. 7 (2010): 1259-1276.

Rosenzweig, Cynthia, Joshua Elliott, Delphine Deryng, Alex C. Ruane, Christoph Müller, Almut Arneth, Kenneth J. Boote et al. "Assessing agricultural risks of climate change in the 21st century in a global gridded crop model intercomparison." *Proceedings of the National Academy of Sciences* 111, no. 9 (2014): 3268-3273.

Calvin, Katherine, Ben Bond-Lamberty, Leon Clarke, James Edmonds, Jiyong Eom, Corinne Hartin, Sonny Kim et al. "The SSP4: A world of deepening inequality." *Global Environmental Change* 42 (2017): 284-296.

Turner, Sean WD, Mohamad Hejazi, Katherine Calvin, Page Kyle, and Sonny Kim. "A pathway of global food supply adaptation in a world with increasingly constrained groundwater." *Science of The Total Environment* 673 (2019): 165-176.

Sağlam, Yiğit. "Pricing of water: optimal departures from the inverse elasticity rule." *Water Resources Research* 49, no. 12 (2013): 7864-7873.

Chaturvedi, Vaibhav, Mohamad Hejazi, James Edmonds, Leon Clarke, Page Kyle, Evan Davies, and Marshall Wise. "Climate mitigation policy implications for global irrigation water demand." *Mitigation and Adaptation Strategies for Global Change* 20, no. 3 (2015): 389-407.

Graham, Neal T., Mohamad I. Hejazi, Min Chen, Evan GR Davies, James A. Edmonds, Son H. Kim, Sean WD Turner et al. "Humans drive future water scarcity changes across all Shared Socioeconomic Pathways." *Environmental Research Letters* 15, no. 1 (2020): 014007.

Popp, A., Calvin, K., Fujimori, S., Havlik, P., Humpenöder, F., Stehfest, E., ... & Hasegawa, T. (2017). Land-use futures in the shared socio-economic pathways. *Global Environmental Change*, 42, 331-345.

Riahi, K., Van Vuuren, D. P., Kriegler, E., Edmonds, J., O'Neill, B. C., Fujimori, S., ... & Lutz, W. (2017). The shared socioeconomic pathways and their energy, land use, and greenhouse gas emissions implications: an overview. *Global Environmental Change*, 42, 153-168.

O'Neill, B. C., Kriegler, E., Ebi, K. L., Kemp-Benedict, E., Riahi, K., Rothman, D. S., ... & Levy, M. (2017). The roads ahead: Narratives for shared socioeconomic pathways describing world futures in the 21st century. *Global Environmental Change*, 42, 169-180.

Fricko, O., Havlik, P., Rogelj, J., Klimont, Z., Gusti, M., Johnson, N., ... & Ermolieva, T. (2017). The marker quantification of the Shared Socioeconomic Pathway 2: A middle-of-the-road scenario for the 21st century. *Global Environmental Change*, 42, 251-267.

Fujimori, S., Hasegawa, T., Masui, T., Takahashi, K., Herran, D. S., Dai, H., ... & Kainuma, M. (2017). SSP3: AIM implementation of shared socioeconomic pathways. *Global Environmental Change*, *42*, 268-283.

Graham, N. T., Davies, E. G., Hejazi, M. I., Calvin, K., Kim, S. H., Helinski, L., ... & Wise, M. A. (2018). Water Sector Assumptions for the Shared Socioeconomic Pathways in an Integrated Modeling Framework. *Water Resources Research*, *54*(9), 6423-6440.

Kriegler, E., Bauer, N., Popp, A., Humpenöder, F., Leimbach, M., Strefler, J., ... & Mouratiadou, I. (2017). Fossil-fueled development (SSP5): an energy and resource intensive scenario for the 21st century. *Global environmental change*, *42*, 297-315.

Van Vuuren, D. P., Stehfest, E., Gernaat, D. E., Doelman, J. C., Van den Berg, M., Harmsen, M., ... & Girod, B. (2017). Energy, land-use and greenhouse gas emissions trajectories under a green growth paradigm. *Global Environmental Change*, *42*, 237-250.

Van Vuuren, Detlef P., Jae Edmonds, Mikiko Kainuma, Keywan Riahi, Allison Thomson, Kathy Hibbard, George C. Hurtt et al. "The representative concentration pathways: an overview." *Climatic change* 109, no. 1-2 (2011): 5.

O'Neill, B. C., Tebaldi, C., Vuuren, D. P. V., Eyring, V., Friedlingstein, P., Hurtt, G., ... & Meehl, G. A. (2016). The scenario model intercomparison project (ScenarioMIP) for CMIP6. *Geoscientific Model Development*, *9*(9), 3461-3482.

Eyring, V., Bony, S., Meehl, G. A., Senior, C. A., Stevens, B., Stouffer, R. J., & Taylor, K. E. (2016). Overview of the Coupled Model Intercomparison Project Phase 6 (CMIP6) experimental design and organization. *Geoscientific Model Development (Online)*, *9*(LLNL-JRNL-736881).

Kriegler, E., Edmonds, J., Hallegatte, S., Ebi, K. L., Kram, T., Riahi, K., ... & Van Vuuren, D. P. (2014). A new scenario framework for climate change research: the concept of shared climate policy assumptions. *Climatic Change*, *122*(3), 401-414.

Turner, S. W., Hejazi, M., Kim, S. H., Clarke, L., & Edmonds, J. (2017). Climate impacts on hydropower and consequences for global electricity supply investment needs. *Energy*, *141*, 2081-2090.

Li, X., Vernon, C. R., Hejazi, M. I., Link, R. P., Feng, L., Liu, Y., & Rauchenstein, L. T. (2017). Xanthos—A Global Hydrologic Model. *Journal of Open Research Software*, *5*(PNNL-SA-126584).

Liu, Yaling, Mohamad Hejazi, Hongyi Li, Xuesong Zhang, and Guoyong Leng. "A hydrological emulator for global applications—HE v1. 0.0." *Geoscientific Model Development* 11, no. 3 (2018): 1077-1092.

Vernon, C. R., Hejazi, M. I., Turner, S. W., Liu, Y., Braun, C. J., Li, X., & Link, R. P. (2019). A Global Hydrologic Framework to Accelerate Scientific Discovery. *Journal of Open Research Software*, 7(1).

Warszawski, L., Frieler, K., Huber, V., Piontek, F., Serdeczny, O., & Schewe, J. (2014). The inter-sectoral impact model intercomparison project (ISI-MIP): project framework. *Proceedings of the National Academy of Sciences*, 111(9), 3228-3232.

Clarke, L., Eom, J., Marten, E. H., Horowitz, R., Kyle, P., Link, R., ... & Zhou, Y. (2018). Effects of long-term climate change on global building energy expenditures. *Energy Economics*, 72, 667-677.

REVIEWERS' COMMENTS:

Reviewer #1 (Remarks to the Author):

I think the authors have addressed my main concerns. The manuscript is now much clearer and the readers will be able to understand how the study has been developed and appreciate the complexity of the analyses proposed by the authors.

I still think this is an important contribution to the literature and I would be excited to see it published in Nature Communications.

Reviewer #2 (Remarks to the Author):

The revised manuscript has properly addressed my comments on the initial submission.